# Effects of κ-Carrageenan and Guar Gum on the Rheological Properties and Microstructure of Phycocyanin Gel

**DOI:** 10.3390/foods11050734

**Published:** 2022-03-02

**Authors:** Yu-chen Lei, Xia Zhao, Dong Li, Li-jun Wang, Yong Wang

**Affiliations:** 1Beijing Key Laboratory of Functional Food from Plant Resources, College of Food Science and Nutritional Engineering, China Agricultural University, P.O. Box 50, 17 Qinghua Donglu, Beijing 100083, China; lyc971121@163.com (Y.-c.L.); zhaoxiaa@cau.edu.cn (X.Z.); 2Beijing Advanced Innovation Center for Food Nutrition and Human Health, National Energy R & D Center for Non-Food Biomass, College of Engineering, China Agricultural University, Beijing 100083, China; dongli@cau.edu.cn; 3School of Chemical Engineering, The University of New South Wales, Sydney, NSW 2052, Australia; yong.wang2@unsw.edu.au

**Keywords:** phycocyanin, κ-carrageenan, guar gum, rheological properties

## Abstract

The effects of two polysaccharides on the performance and microstructure of phycocyanin gels were studied by choosing anionic polysaccharides (κ-carrageenan) and neutral polysaccharides (guar gum). The linear and nonlinear rheological properties and microstructure of the phycocyanin-polysaccharide composite gel were evaluated. The results show that both κ-carrageenan and guar gum can enhance the network structure of phycocyanin gel and weaken the frequency dependence. The sample with 0.4% κ-carrageenan has the highest gel strength. All samples exhibited Type I behavior (inter-cycling strain-thinning) and mainly elastic behavior. As the concentration of κ-carrageenan increases, hydrophobic interactions and disulfide bonds play an essential role in maintaining the three-dimensional structure of the gel. Too high a concentration of guar gum hinders the formation of protein disulfide bonds. This research can provide a theoretical basis for designing and developing new food products based on phycocyanin and different polysaccharides with ideal texture in the food industry.

## 1. Introduction

Proteins and polysaccharides are two essential biological macromolecules and important components in food systems [1]. Proteins generally have polar and non-polar groups, in order to have good emulsification and foaming properties. Further, proteins have good solubility and gelation due to protein intermolecular interaction [2]. Different polysaccharides have different types of chemical structures, thus they can be used as thickeners, gelling agents, and stabilizers [3]. Therefore, they can be used as natural food additives and play a vital role in changing food taste, structure, texture, and stability [4,5,6,7].

κ-carrageenan is extracted from edible red algae. It is an anionic polysaccharide, a linear sulfated polysaccharide formed by D-galactose and 3,6-anhydro-D-galactose [8]. According to the number and location of sulfuric acid groups, carrageenan is generally divided into three types: κ-, ί-, and λ-carrageenan [9]. As the most commonly used type, κ-carrageenan (CG) is an excellent thickener and gelling agent, and is often used to change viscoelasticity [10,11]. Guar gum (GG) is extracted from the endosperm part of the seeds of the guar plant. It is a neutral polysaccharide [12]. The guar gum molecule consists of linear α-1,4-linked mannose units and randomly linked β-1,6-linked galactose unit composition [13]. In guar gum, the ratio of mannose/galactose (M:G) is about 1.55 [14]. In the food industry, guar gum is used in various foods, such as sauces, soups, dairy products, and baked goods.

Phycocyanin is the main protein in spirulina [15]. It has a bright blue color and can be used in food as a pigment [16]. In addition to being used as pigment, phycocyanin also has a series of biological functions such as good antioxidant, anti-inflammatory, anti-tumor, improving immunity, preventing organ damage, etc. [17]. It has been used as a pigment in desserts, beverages, ice cream, and other foods, as well as having certain applications in the fields of health food, cosmetics, and biomedicine [18]. However, the current research on the rheological properties of phycocyanin-polysaccharide gel is still blank.

Rheological properties can provide valuable information for understanding the microstructure of complex fluids [19,20,21]. In recent years, large amplitude oscillatory shear (LAOS) has received more attention, especially in rheological analysis of gels, suspensions, or colloids [22]. This is due to the theory of linear viscoelasticity being valid only when the total deformation is very small. However, in the food processing industry, the deformation of materials is generally large and rapidly reaches the nonlinear region. At this time, using the fluid characteristics obtained in the linear region to describe the flow characteristics of the material is not enough, and the nonlinear rheological characteristics are closer in mimicking the reality [23]. Complex fluids with similar linear rheological properties in the linear viscoelastic region (LVR) may exhibit different nonlinear rheological properties in the nonlinear viscoelastic region. This means that the LAOS test can better distinguish the structural differences of soft materials [22]. In addition, compared with the traditional small-amplitude oscillating shear test that does not cause permanent deformation of food materials, the large-scale oscillating shear test can better simulate the process of food being chewed in the mouth [24].

At present, the rheological properties and microstructure of protein-polysaccharide gels have been extensively studied [25,26,27,28,29,30,31,32,33], but there was little research on the nonlinear rheological properties of phycocyanin and phycocyanin-polysaccharide gels [34]. Therefore, this article focuses on phycocyanin-polysaccharide gel, which is of great significance for studying nonlinear rheological properties and microstructure.

We hypothesize that polysaccharides affect the gel strength of phycocyanin, and different types and concentrations of polysaccharides influence the rheological properties and microstructure of phycocyanin gels. The purpose of this study is to explore the effects of different κ-carrageenan and guar gum additions on the rheological properties and microstructure of phycocyanin gels. Therefore, this study can contribute to developing new foods containing phycocyanin by revealing the interaction mechanism between phycocyanin and κ-carrageenan and guar gum.

## 2. Materials and Methods

### 2.1. Materials

Phycocyanin (PC) was obtained from Xinyuli Algae Industry Group Co., Ltd. (Ordos, China). Guar gum (GG, the molecular weight is about 50,000–80,0000) and κ-carrageenan (CG, the molecular weight is about 8000–10,5000) were acquired from Source Leaf Biotechnology Co. Ltd (Beijing, China). Sodium hydroxide (NaOH), Tris, glycine, Na_2_EDTA·2H_2_O, sodium dodecyl sulfate (SDS), urea, β-mercaptoethanol, rhodamine B, and other chemicals were used in analytical grade. Potassium Bromide (KBr) was spectrum pure. All chemicals were supplied by Shanghai Macklin Biochemical Co., Ltd. (Shanghai, China). Deionized water was used for all experiments. The solvents used for the gel solubility test are as shown below: distilled water (pH = 8.0, adjusted by 0.5 M NaOH) (S_1_); 0.086 mol/L Tris-0.09 M Glycine-4 mmol/L Na_2_EDTA (pH = 8.0) (S_2_); S_2_ with 0.5% SDS (S_3_); S_3_ with 8 mol/L Urea (S_4_); S_4_ with 2% 2-mercaptoethanol (S_5_).

### 2.2. Sample Preparation

The CG or GG powder were dispersed in the deionized water, stirring at 1000 rpm for 4 h at room temperature (25 °C) to get the CG or GG stock solutions (0.4%, *w*/*v*). The stock solutions were diluted to prepare 0.1, 0.2, 0.4% *w*/*v* CG or GG solutions. A certain amount of PC was added into the CG or GG solution, stirring at 1000 rpm for 4 h at room temperature (25 °C) to make the concentration of PC 16%, *w*/*v*. The mixtures of PC (16%, *w*/*v*) and CG or GG (0.1, 0.2, 0.4%, *w*/*v*) were kept overnight at 4 °C for complete hydration. The samples were adjusted to pH 7.0 using 0.5 M NaOH solution. The samples were heated in a water bath at 90 °C for 30 min [35] and cooled in an ice-water bath for 10 min to prepare PC-CG and PC-GG gel samples.

### 2.3. Rheological Properties

The rheological properties of PC-CG and PC-GG gels were measured using an AR2000ex rheometer (TA, Crawley, UK) with an aluminum parallel plate geometry (40 mm diameter, 1000 μm gap). We refer to the previous study [36] with slight changes. For each test, the mixture of PC-GG and PC-CG needed to be in equilibrium on the Peltier plate at 25 °C for 2 min. The mixtures were heated from 25 °C to 90 °C at a rate of 10 °C/min, held at 90 °C for 30 min, and then cooled down to 25 °C at 20 °C/min. The samples were equilibrated at 25 °C for 5 min to get PC-CG or PC-GG in-situ thermal gel for future tests. A thin layer of silicone oil (350 ± 8 mPa·s) was used on the edge of the samples to prevent evaporation.

#### 2.3.1. Small Amplitude Oscillatory Shear (SAOS) Testing

The frequency sweeps were used for SAOS tests. The tests were carried out at 25 °C from 0.1 to 10 Hz at a constant strain of 0.5% (within the LVR). The storage modulus (*G*′) and loss modulus (*G*″) were obtained. The frequency dependency of the storage (*G*′) and loss (*G*″) moduli were approximated by Equations (1) and (2) [37].
(1)$$G^{\prime }=K^{\prime }\times \omega^{n\prime }$$
(2)$$G^{\prime \prime }=K^{\prime \prime }\times \omega^{n\prime \prime }$$
where *K*′ and *K*″ are the power-law model constants (Pa/s^n^), *n*′ and *n*″ stand for the frequency exponents (no dimensionless), and *ω* means the angular frequency (rad·s^−1^).

#### 2.3.2. Large Amplitude Oscillatory Shear (LAOS) Testing

The strain sweeps were used for LAOS tests. Strain sweep tests were carried out at 25 °C from 0.1 to 100% at 1 Hz. The critical strain amplitude is the strain level where *G*′ dropped by 5% in the LVR [38]. The critical strain (*γ_c_*) and the corresponding storage modulus (*G_cr_*) were obtained through the experiments. The cohesive energy density (*E_c_*) could be calculated by the following Equation (3) [39].
(3)$$E_{c}=\frac{1}{2}\gamma_{c}^{2}G_{cr}$$
where *γ_c_* means the critical strain, *G_cr_* means the corresponding storage modulus, and *E_c_* means the cohesive energy density. 

In addition, the MITlaos program (MITlaos beta) was used to collect the original stress and strain data to analyze the Lissajous curve, the third harmonic viscoelastic moduli, and the Chebyshev coefficient.

### 2.4. Gel Solubility

In order to explore the non-covalent and covalent interactions involved in the PC-CG and PC-GG thermal gels, the solubility of gels was measured in different buffers according to the method described by [40] with slight modifications. A certain amount of gel sample (protein concentration of 8 mg/mL) was taken and dissolved in five different solvents (S_1_–S_5_, mentioned in Section 2.1) with the same volume. A certain amount of gel samples were mixed with five different solvents, shaken for 1 min, kept at 25 °C for 15 min, and then centrifuged at 8000 rpm for 15 min. The supernatant of the proteins was determined according to [41] method using a UV-vis spectrometer (TU1810, Beijing Puxi General Instrument Co., Ltd., Beijing, China) at a wavelength of 750 nm. The bovine serum albumin was used as the standard. The gel solubility was expressed as the ratio of the supernatant protein content to total protein content [6].

### 2.5. Fourier Transform Infrared Spectroscopy (FTIR)

The functional groups of samples were measured by a Fourier Transform Infrared Spectroscopy (Spectrum 100, Perkin-Elmer Co., Waltham, MA, USA). The sample and potassium bromide were fully ground, mixed thoroughly, and then placed in the sample holder after being compressed into thin slices. The measurements were carried out in the scanning range from 4000 cm^−1^ to 400 cm^−1^ with a resolution of 4 cm^−1^ in 32 scans.

### 2.6. Water Holding Capacity (WHC)

The WHC of the hybrid gel was determined according to the method described by [42], but with slight modifications. The mixed gel of 5 g was centrifuged at 6000 rpm for 10 min at 4 °C. The supernatant was discarded after centrifugation. The differences of the samples quality were recorded before and after centrifugation. The water holding capacity is calculated by the following Equation (4).
(4)$$\mathrm{WHC}=\frac{\mathrm{Sample}\;\mathrm{weight}\;\mathrm{after}\;\mathrm{centrifugation}}{\mathrm{Sample}\;\mathrm{weight}\;\mathrm{before}\;\mathrm{centrifugation}}\text{×}100\%$$

### 2.7. Confocal Laser Scanning Microscopy (CLSM)

The Nikon confocal laser scanning microscope (A1, Nikon Solutions Co., Ltd., Tokyo, Japan) equipped with an inverted microscope was used to observed the spatial network distribution of the samples. Rhodamine B (0.1%, *w*/*v*) was used to stain the protein in the sample according to the previously reported method [25]. The samples were stained with the fluorescent dye in a volume ratio of 1:50 before gelation. The mixture was added into the grooved glass slide, covered with the cover glass and sealed. The samples were heated in a water bath at 90 °C for 30 min, and were cooled to room temperature as soon as possible. Then observed with 60 × objective lens and 10 × eyepiece. Rhodamine B was excited at 543 nm and the reception wavelength was 561–700 nm. The pre-installed image processing software (NIS Viewer 5.21, Nikon Instruments Inc., Shanghai, China) was used to acquire images (1024 × 1024 pixels per image).

### 2.8. Statistical Analysis

All the experimental results were repeated at least triplicate to calculate the means ± standard deviation. SPSS software (SPSS 26, IBM SPSS Statistics, Chicago, IL, USA) and MATLAB software (MATLAB R2014a, Natick, MA, USA) were used for statistical analysis. Differences were considered statistically significant (*p* < 0.05) by the one-way analysis of variance (ANOVA) with Duncan’s test. Besides, Origin 2018 (Originlab Inc., Northampton, MA, USA) was used to draw figures of all tests.

## 3. Results and Discussion

### 3.1. Rheology Properties

#### 3.1.1. Linear Rheology Properties

The PC-CG and PC-GG samples were analyzed with frequency sweep tests (Figure 1A_1_,A_2_). As shown in Figure 1, all samples, *G*′ values were greater than *G*″ values, indicating that all samples exhibit a gel-like structure. The *G*′ and *G*″ values of PC-CG hybrid gel and PC-GG hybrid gel increased when the frequency increased from 0.1 Hz to 10 Hz, which means that all samples have a certain frequency dependence. For *G*′ and *G*″, the slope of the PC-CG hybrid gel was lower than that of the PC-GG hybrid gel, which shows that the gel network structure of the PC-CG hybrid gel was less susceptible to frequency. In the PC-CG and PC-GG hybrid gels, as CG and GG concentrations increased, both *G*′ and *G*″ values increased. Cavallieri et al. [43] found that addition of κ-carrageenan could increase the *G*′ and *G*″ of heat induced soy protein isolate gels, which is similar to the test results of PC-CG gels. This is due to the synergistic effect of phycocyanin and κ-carrageenan, which promotes the formation of a dense gel structure. In PC-GG gels, the *G*′ and *G*″ of all samples were larger than those without GG. The samples with 0.1% GG had the larger *G*′ and *G*″ in the tested range, while the samples with 0.2% and 0.4% GG had lower *G*′ and *G*″ than the samples with 0.1% GG. This result is similar to the study by [44]. It shows that when the concentration of guar gum is high, the entangled galactomannan chains hamper the interaction between proteins [44].

In order to further evaluate the gels’ strength, the frequency (*ω*) dependence of *G*′ and *G*″ of the gels was fitted by the Power-law model. The fitted data are shown in Table 1. For all samples, the Power-law model of storage modulus *G*′ and loss modulus *G*″ fitted well (*R*^2^ > 0.966). For the samples added with κ-carrageenan, the values of *K*′ and *K*″ were positively associated with κ-carrageenan concentration. The increase of *K*′ and *K*″ indicated that the elasticity and viscosity of PC-CG gels were enhanced. The values of *n*′ and *n*″ were negatively related to the κ-carrageenan concentration, indicating that the frequency-dependent sensitivity of phycocyanin-κ-carrageenan gel decreased with the increase of the amount of κ-carrageenan. For the samples with guar gum, the values of *K*′ and *K*″ peaked when the guar gum was added at 0.1%, and then gradually decreased. The values of *n*′ and *n*″ were not affected by the amount of guar gum, and both were lower than the control sample. It shows that the κ-carrageenan and guar gum can both reduce the frequency dependence sensitivity of phycocyanin thermal gel, and the effect of κ-carrageenan is better than guar gum at the same addition.

#### 3.1.2. Nonlinear Rheological Properties

##### LAOS Behaviors Analysis

The LAOS behaviors of the PC-CG and PC-GG gels at various polysaccharide concentrations were shown in Figure 1B_1_,B_2_. In the LVR, *G*′ was far larger than *G*″ of all samples, indicating a dominant solid-like behavior or elastic response. In addition, all the samples’ *G*′ and *G*″ values remained stable in the linear viscoelastic region, regardless of the strain amplitude. 

According to the performance of modulus in the strain range, the rheological behavior of a complex fluid can be divided into at least four different types in the nonlinear region [22]. *G*′ and *G*″ values dropped remarkably after the *γ_c_* with the strain increase for all the samples. Therefore, PC-CG gel and PC-GG gel belong to Type I (inter-cycling strain-thinning) network [45]. A similar result has been reported in soy protein isolate-locust bean gum acid-induced gel [25] and pea protein thermal gel [46].

Generally, *G*′ can represent the hardness of the gel. In order to further compare the effects of different kinds of polysaccharides on the gel hardness, the *γ_c_*, *G_cr_*, and the *E_c_* were calculated [47] and presented in Table 2. The *E_c_* is an effective indication to characterize the attractive interaction between molecules and assess the cohesive ability of substances [48]. Higher cohesive energy density indicates higher structural strength [49]. The addition of two polysaccharides had a significant effect on gel hardness. Either the different kinds of polysaccharides or the different addition of polysaccharides could significantly increase the *γ_c_* of the gel. When the κ-carrageenan concentration was 0.4%, the gel had a larger critical strain (1663 Pa for PC-CG gels), indicating it existed a long LVR and could maintain similar properties under large deformations. The *G_cr_* and *E_c_* under *γ_c_* of the sample with 0.4% CG were significantly higher than other samples, which is consistent with the results of frequency sweep tests and strain sweep tests and further expressing the strength of the gel [50].

##### Lissajous Plots Analysis

A more detailed assessment of the nonlinear viscoelastic behavior of PC-CG and PC-GG gels was obtained using Lissajous curves. The advantage of this method is that the nonlinear signal obtained in the LAOS test is decomposed into parts corresponding to elasticity and viscosity, without the need to apply any constitutive equations [51]. As shown in Figure 2, the shape and area of the Lissajous curves can provide information about the change in the viscoelasticity of the material as the strain amplitude increases. In the LVR, the shapes of the elastic Lissajous curves of all samples were elliptical since both stress and strain were sinusoidal waves. As the phycocyanin gels entered the nonlinear region, the shape of the elastic Lissajous curve of all samples changed from an ellipse to a parallelogram, indicating plastic behavior [52]. When the strain was larger than 50%, the elastic Lissajous curves of all samples deviated from the ellipse. With the increase of the strain amplitude, the areas of the elastic Lissajous curves of all samples became larger, indicating the fracture of the gel structure [53]. The viscous Lissajous curves of 16% PC gel were ellipsoidal (viscoelastic) at the range of strain 1–50%, followed by self-intersection points when strain rose to 100%. The self-intersection points appeared in the viscous Lissajous curves at a large strain rate, which may be related to the strong nonlinearity in the elastic stress [54]. These self-intersections may occur when the time scale for restructuring of the microstructure is shorter than the oscillatory deformation time scale [55]. The areas of the elastic Lissajous curves of all samples were smaller than the viscous Lissajous curves all along, showing the elastic-dominated behavior. Furthermore, the area of the elastic Lissajous curve increased as the concentration of CG or GG increased, suggesting that the addition of CG and GG enhanced the viscosity of the gel at the nonlinear viscoelastic regime [56]. The ellipsoidal shape of their elastic Lissajous curves evolved to be more spherical at high strains, resulting from shear-thinning behavior.

##### Fourier-Transform Rheology Analysis

The Fourier Transform method was used to decompose the stress response into higher harmonics to quantify the LAOS behavior. The nonlinear stress response of materials can be quantified by the relative intensity of high harmonics (*I_n_*/*I*_1_), and provides some information about the microstructure or molecular structure of the materials. The stress response was no longer in a sine waveform at the large strain amplitude. The stress response could no longer be fully described in the storage modulus (*G*′) and loss modulus (*G*″). This is due to the high harmonic contribution, especially the third harmonic contribution [22]. In this study, we chose to focus on the change of *I*_3_/*I*_1_ to discuss the influence of higher harmonics on the rheological behavior of phycocyanin gel. The maximum strength of *I*_3_/*I*_1_ may be related to the large aggregate and structural breakdown under high strain amplitudes. The effects of different additions of CG and GG on the relative strength (*I*_3_/*I*_1_) of the third harmonic of phycocyanin gel were shown in Figure 3A_1_,A_2_, respectively. It can be seen from the figure that the *I*_3_/*I*_1_ of all samples increased with the increase of strain amplitude, when the *I*_3_/*I*_1_ increased by 5%, the corresponding strain is about 20%. It is indicated that an increased influence of the third harmonic on the stress response in the nonlinear region. The third harmonic appearance indicated the system’s transition from the linear viscoelastic range into the nonlinear viscoelastic range. This is since the sample was subjected to time-varying deformation and had undergone intense deformation, and the changes in their structure appear as an additional third harmonic component [57]. The maximum strength of *I*_3_/*I*_1_ may be related to the alignment of large aggregates or their break up at very high strain amplitudes [58].

##### Chebyshev Coefficients Analysis

The Chebyshev stress decomposition method can be used to analyze the viscoelastic properties of samples in the nonlinear viscoelastic region. The total stress response can be decomposed into the elastic part and viscous parts. Elastic and viscous behavior can be explained by discrete Chebyshev harmonic coefficients [59]. In the large amplitude deformation, the third Chebyshev coefficients (*e*_3_ and *v*_3_) were mainly used to predict the nonlinear response and provide some information about the elastic and viscous components. Therefore, the physical interpretation of the Fourier coefficients (third harmonic) can be explained by the signs of *e*_3_/*e*_1_ and *v*_3_/*v*_1_ as: strain-stiffening (*e*_3_/*e*_1_ > 0), linear elasticity (*e*_3_/*e*_1_ = 0), strain-softening (*e*_3_/*e*_1_ < 0); shear-thickening (*v*_3_/*v*_1_ > 0), linear viscous (*v*_3_/*v*_1_ = 0) and shear-thinning (*v*_3_/*v*_1_ < 0) [60]. 

As shown in Figure 3B_1_,B_2_, the *e*_3_/*e*_1_ of PC-CG and PC-GG gels initially remained close to zero, *e*_3_/*e*_1_ increased slowly as the strain increased, when *e*_3_/*e*_1_ is increased by 5%, the corresponding strain is about 20%. This means a shift from linear elastic behavior to strain-stiffening behavior. The strain-stiffening behavior was driven by the third nonlinear harmonic [59]. As shown in Figure 3B_1_, as the concentration of κ-carrageenan increased, strain-stiffening became increasingly clear, indicating the significant contribution of the third harmonic, which was also in agreement with the results of higher harmonic analysis. The increasing strain-stiffening behavior showed an enhanced resistance to deformation of the PC-CG gel [61]. This trend could be explained by the phase separation between κ-carrageenan and phycocyanin, which increased the concentration of phycocyanin in the continuous phase [62]. As illustrated in Figure 3B_2_, PC-GG gel showed the strongest strain-stiffening behavior when the guar gum was 0.1%. As the concentration of guar gum increased, the strain-stiffening behavior gradually weakened. This may be due to the increased aggregation of denatured protein molecules and the increased gel strength under the action of low concentration of guar gum; higher concentrations of guar gum caused excessive aggregation (incipient precipitation), resulting in the decreasing of gel strength. As presented in Figure 3C_1,_C_2_, the *v*_3_/*v*_1_ (third-order viscosity Chebyshev coefficient ratio) of PC-CG and PC-GG gels fluctuated around 0 at first, then dropped sharply around 10%, and around 50% reached the lowest point and then gradually raised. The *v*_3_/*v*_1_ show 10–15% change at about 20% strain, which means significant viscous contribution in all samples. This indicated that all samples have a linear viscosity to shear-thinning change as the strain amplitude increased.

### 3.2. Chemical Interaction Forces Analysis

The solubility of PC-CG and PC-GG gels in five different solvents were measured to clarify the role of protein-polysaccharide interaction in forming the gel network. The differences between two adjacent solvents represented electrostatic interaction (S_2_-S_1_), hydrophobic interaction (S_-_S_2_), hydrogen bond (S_4_-S_3_), and disulfide bond (S_5_-S_4_) [6]. Figure 4A,B, respectively, showed the effect of κ-carrageenan (A) and guar gum (B) on the interaction between phycocyanin gel molecules. Figure 4A showed that when the concentration of κ-carrageenan increased, electrostatic interactions and hydrogen bond interactions gradually decreased. At the same time, hydrophobic interactions and disulfide bond interactions increased significantly with the increase of κ-carrageenan concentration (*p* < 0.05). The interaction force between phycocyanin and κ-carrageenan was disulfide bond interaction > hydrophobic interaction > electrostatic interaction > hydrogen bond interaction. This phenomenon indicated that hydrophobic interactions and disulfide bond interactions play a vital role in forming PC-CG gels. It may be due to the increase in temperature during the heating process that physically induced hydrophobic interactions in the protein-polysaccharide gel. During the heating process, the biopolymers underwent conformational and structural changes, and contacted and interacted with the hydrophobic part of the biopolymers [63]. The conformational changes of the protein during the heat treatment led to an increase in the interaction between the protein and other substances [64]. κ-carrageenan acted as a cross-linking agent in the composite gel. More hydrophobic groups and sulfhydryl groups are exposed by heating to denature the protein. At this time, hydrophobic interactions occurred in the system, which caused the aggregation of protein molecules, and finally promoted the gel structure. The exposure of sulfhydryl groups and the addition of κ-carrageenan were both beneficial to the formation of disulfide bonds in the gel [65]. Figure 4B demonstrated that when the concentration of guar gum increased, electrostatic interactions and hydrophobic interactions were slightly increased. The hydrogen bond reached the maximum when the guar gum’s concentration was 0.1%, and then gradually decreased. The disulfide bond interaction decreased with the increase of guar gum concentration. This may be due to the increased degree of aggregation of denatured protein molecules and increased gel strength under the action of low concentration of guar gum, at high concentrations, guar gum occupied a larger space volume, which hindered the formation of the disulfide bonds. In addition, [44] found that a lower concentration of guar gum could increase the denaturation temperature of whey protein isolate. At the same time, a higher concentration of guar gum would lower the denaturation temperature of whey protein isolate. The insufficient exposure of sulfhydryl groups reduced disulfide bond content, which may be used to explain the decrease of sulfhydryl groups in PC-GG gels in this article.

### 3.3. FTIR Spectroscopy Analysis

Fourier transform infrared spectroscopy (FT-IR) was used to further detect the structural changes of proteins and the interaction between protein-polysaccharides. As shown from Figure 5, the FT-IR spectra of all gel samples were similar, indicating that the functional groups were found during the gel formation process, despite the addition of different concentrations of κ-carrageenan or guar gum in the gel did not change significantly. The samples showed typical absorption peaks in different wavelength ranges, representing different functional groups. Specifically, the peak at 3200–3400 cm^−1^ was the extension of hydroxyl hydration, indicating the existence of hydrogen bond interactions [28]. The amide I band at 1600–1700 cm^−1^ indicated C=O stretching, which can be used to analyze the secondary structure of proteins [66]. The band at 800–1200 cm^−1^ was related to the extension of C-O and C-C and the bending of C-H [7]. The FT-IR spectrum showed that with the increase of κ-carrageenan or guar gum, the extension of gel hydroxyl hydration gradually shifted from 3337 cm^−1^ to about 3309 cm^−1^, indicating that the hydrogen bonding force in the composite gel had changed [66]. There was no significant difference in the infrared spectrum of the composite gels at the amide I band at 1600–1700 cm^−1^. The composite gels’ secondary structure was analyzed by Fourier self-deconvolution, and the results are shown in Table 3. It could be found that the relative content of α-helix in other samples was lower than the control sample, except for the sample with 0.4% guar gum. For β-sheets, the sample added with 0.4% guar gum was significantly lower than the other samples. This was similar to the rheological results of the gels. [67] also reached a similar conclusion, which may be due to the relative content of the β-sheet structure being positively correlated with the hardness of the gel. In contrast, the relative content of the α-helical structure is negatively correlated with the gel strength [68].

### 3.4. Water Holding Capacity (WHC)

Water holding capacity (WHC) refers to the ability of the protein to retain water and can be used to assess the quality of different gels. The water holding capacity of the gel samples added with different concentrations of κ-carrageenan/guar gum were shown in Figure 6. It can be seen from the figure that the water holding capacities of all gel samples were above 84%, indicating that all samples had good water holding capacity. For the samples added with κ-carrageenan, the water holding capacity of PC-CG gels gradually increased with the increase of κ-carrageenan concentration, and it was 98.9% when the added amount was 0.4%. This is similar to the result of [69]. The WHC of the gel samples is related to the microstructure, with denser gel structures leading to higher WHC. This is consistent with our CLSM results. For the samples added with guar gum, the water holding capacity of PC-GG gels gradually decreased as the concentration of guar gum increased, and it was 84.5% when the added amount was 0.4%. We can conjecture that the decrease of WHC is due to the phenomenon of saturations caused by polysaccharides concentration [70]. For the amount of polysaccharides added, there may be a concentration threshold. As the polysaccharide concentration increased, the gel became denser, the porosity became smaller, thus the water holding capacity increased. Beyond this threshold, polysaccharides destroyed the microstructure of the gel, the gel became looser and porous, thus the water holding capacity decreased [70]. 

### 3.5. Confocal Laser Scanning Microscope (CLSM)

Figure 7A showed the microstructure of the PC gel, the PC-CG gel, and PC-GG gel under the confocal laser scanning microscope. Only phycocyanin in the sample can be stained by rhodamine B; therefore, the brighter area (red) in the image indicated that the area was rich in phycocyanin. In contrast, the dark area indicated that the area was rich in polysaccharides or water. It can be seen from the figure that the gel network containing only phycocyanin was relatively loose, with more black holes, showing a relatively loose network structure. The pores gradually became less and brighter with the increase of κ-carrageenan concentration, indicating that the samples added with κ-carrageenan had a denser network structure, which was consistent with the results of rheology and water holding capacity. This may be due to carrageenan and phycocyanin filling the originally loose voids and forming a dense network structure [71]. As the addition of guar gum increased from 0% to 0.4%, the pores of the PC-GG gels increased from around 156 μm^2^ to around 625 μm^2^. The pores of the gel tended to increase with the increase of the concentration of guar gum, which can explain the decrease in the water holding capacity of the PC-GG gels with the increase of guar gum. This may be due to the phase separation between phycocyanin and guar gum, which was similar to the results of the study by [72]. The phase separation caused by the lower concentration of guar gum (0.1%) could enhance the aggregation of phycocyanin in the continuous phase, and the storage modulus of the gel gradually increased. The phase separation caused by the higher concentration of guar gum (0.2%, 0.4%) destroyed the continuous phase of the protein, and the storage modulus of the gel gradually decreased. Combining the interaction force and the results of CLSM, we drew the schematic diagram of the microstructure of the PC-CG gel and PC-GG gel, as shown in Figure 7B.

## 4. Conclusions

The effects of κ-carrageenan and guar gum on the linear rheology, nonlinear rheology, and microstructure of phycocyanin gel were investigated. The linear rheological experiments showed that both κ-carrageenan and guar gum could enhance the network structure of phycocyanin gel and weaken the frequency dependence. The sample with 0.4% κ-carrageenan had the greatest gel strength in our tests. The nonlinear rheological experiments showed that all samples exhibited a Type I behavior (inter-cycling strain-thinning). The Lissajous curves showed that the areas of the elastic Lissajous curves of all samples were always smaller than the viscous Lissajous curves, showing a behavior dominated by elasticity. The addition of κ-carrageenan and guar gum enhanced the viscosity of the gel under large strain. The *I*_3_/*I*_1_ of all samples increased with the increase of strain, showing a strain-stiffening behavior. The changes of *e*_3_/*e*_1_ values of all samples with strain amplitude indicated the strain-stiffening behavior. All samples’ *v*_3_/*v*_1_ values indicated a transform from linear viscosity to shear-thinning behavior at large strain amplitudes. The hydrophobic interactions and the disulfide bonds played an essential role in maintaining the three-dimensional structure of the gels as the concentration of κ-carrageenan increased. But an excessively high concentration of guar gum would hinder the formation of protein disulfide bonds. The secondary structure of the gels revealed that the relative content of α-helix in the gel sample with 0.4% guar gum was higher than that of the control sample, while the relative content of β-sheet was lower than that of the control sample. The addition of κ-carrageenan improved the water holding capacity of the gels, while the addition of guar gum reduced the water holding capacity of the gels. The results of confocal laser scanning microscope confirm the results of water holding capacity. In summary, this study may provide a theoretical foundation to design and develop a new edible product based on phycocyanin and different polysaccharides with ideal texture in the food industry. 

## Figures and Tables

**Figure 1 foods-11-00734-f001:**
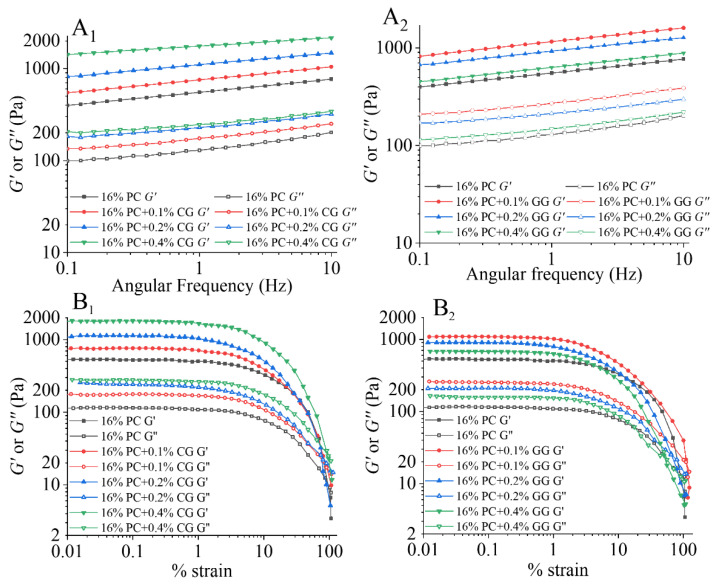
The storage and loss modulus (*G*′ and *G*″) values as increasing of the angular frequency of the PC-CG (**A_1_**) and PC-GG (**A_2_**) gels at various polysaccharide concentrations (CG or GG, 0%-0.4%). The storage and loss modulus (*G*′ and *G*″) values as increasing of the strain of the PC-CG (**B_1_**) and PC-GG (**B_2_**) gels at various polysaccharide concentrations.

**Figure 2 foods-11-00734-f002:**
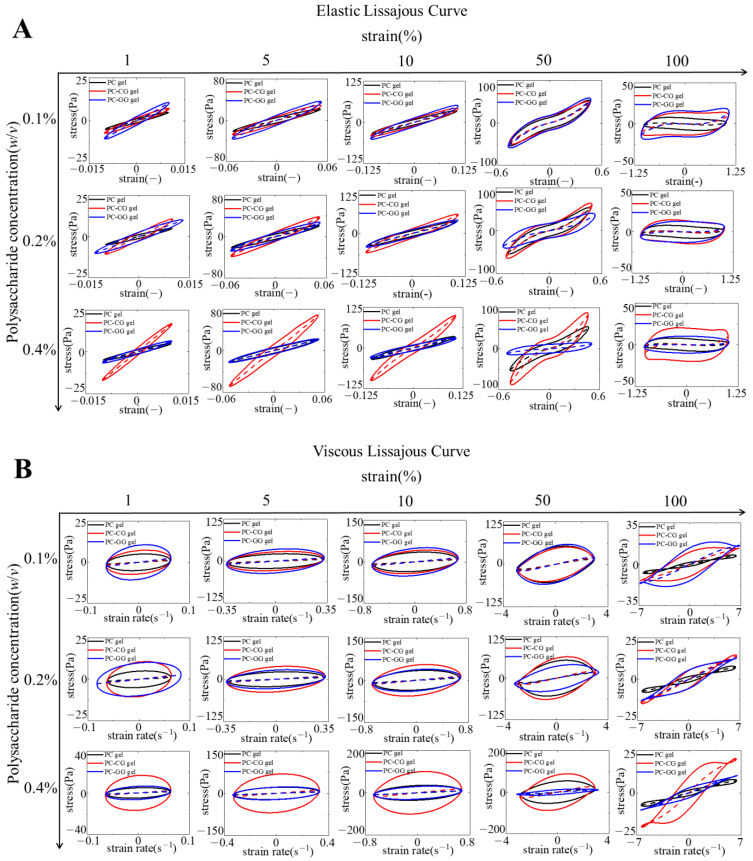
When the strain amplitudes are 1%, 5%, 10%, 50%, and 100%, the elastic Lissajous curve (**A**) and the viscous Lissajous curve (**B**) of 16% PC thermal gels, PC-CG gels, and PC-GG gels, where the black line represents 16% PC thermal gels, the red line represents PC-CG gels, and the blue line represents PC-GG gels. The solid lines represent the total stress, and the dashed lines represent the elastic stress component (**A**) and the viscous stress component (**B**).

**Figure 3 foods-11-00734-f003:**
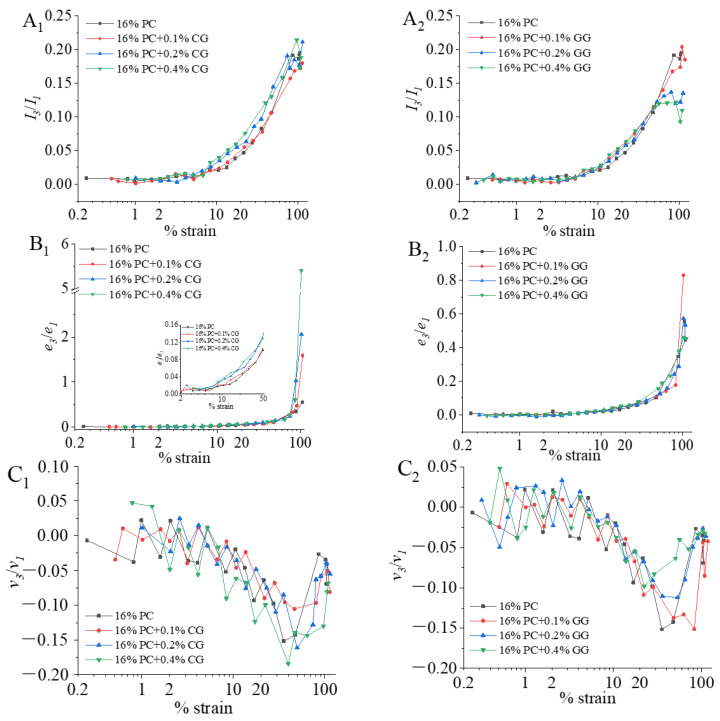
The effect of addition of κ-carrageenan (CG) on *I*_3_/*I*_1_ (**A_1_**), *e*_3_/*e*_1_ (**B_1_**), and *v*_3_/*v*_1_ (**C_1_**) as a function of strain amplitude for the phycocyanin gel (PC); The effect of addition of guar gum (GG) on *I*_3_/*I*_1_ (**A_2_**), *e*_3_/*e*_1_ (**B_2_**), and *v*_3_/*v*_1_ (**C_2_**) as a function of strain amplitude for the phycocyanin gel (PC).

**Figure 4 foods-11-00734-f004:**
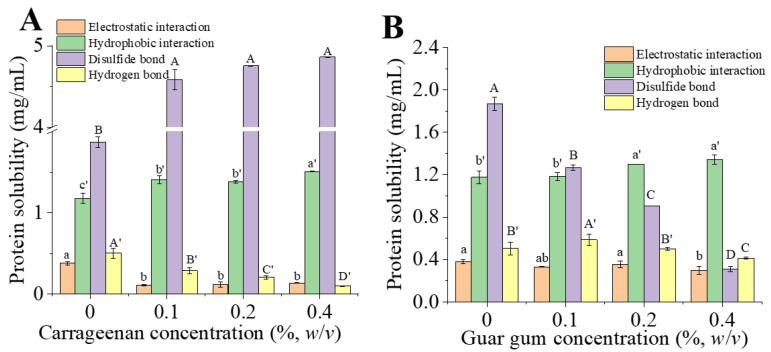
The molecular forces of PC-CG gel (**A**) and PC-GG gel (**B**) at different concentrations of CG/GG. The protein solubility of gels represented the corresponding molecular force. The phycocyanin gel without any polysaccharide was used as a control. Each value represents the mean ± standard deviation (*n* = 3). The values of different letters in the same column were significantly different (*p* < 0.05).

**Figure 5 foods-11-00734-f005:**
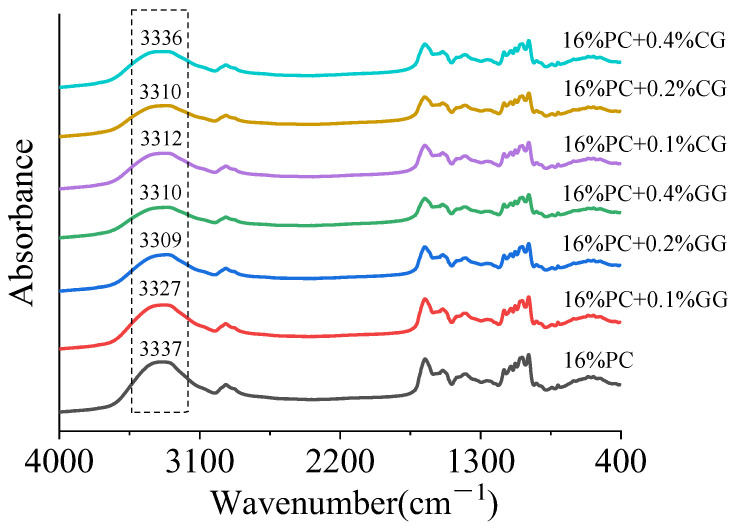
FT-IR spectra of phycocyanin gels with different concentrations of κ-carrageenan/guar gum.

**Figure 6 foods-11-00734-f006:**
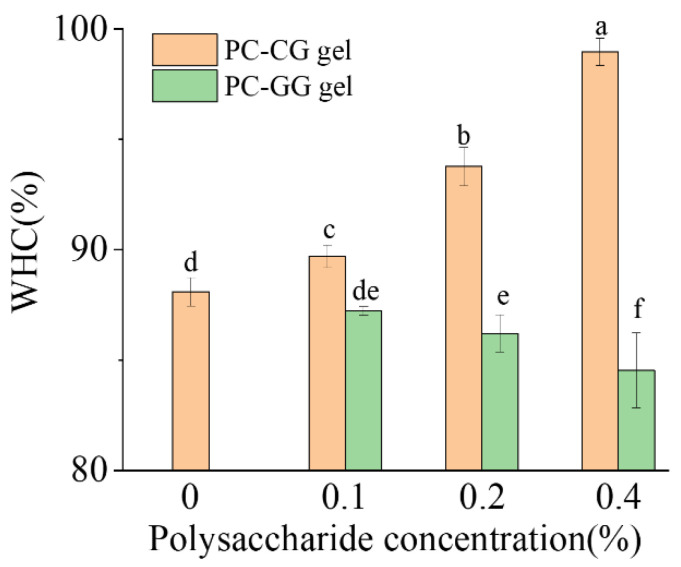
The effect of different κ-carrageenan/guar gum addition on the water holding capacity of phycocyanin gel. Each value represents the mean ± standard deviation (n = 3). The values of different letters were significantly different (*p* < 0.05).

**Figure 7 foods-11-00734-f007:**
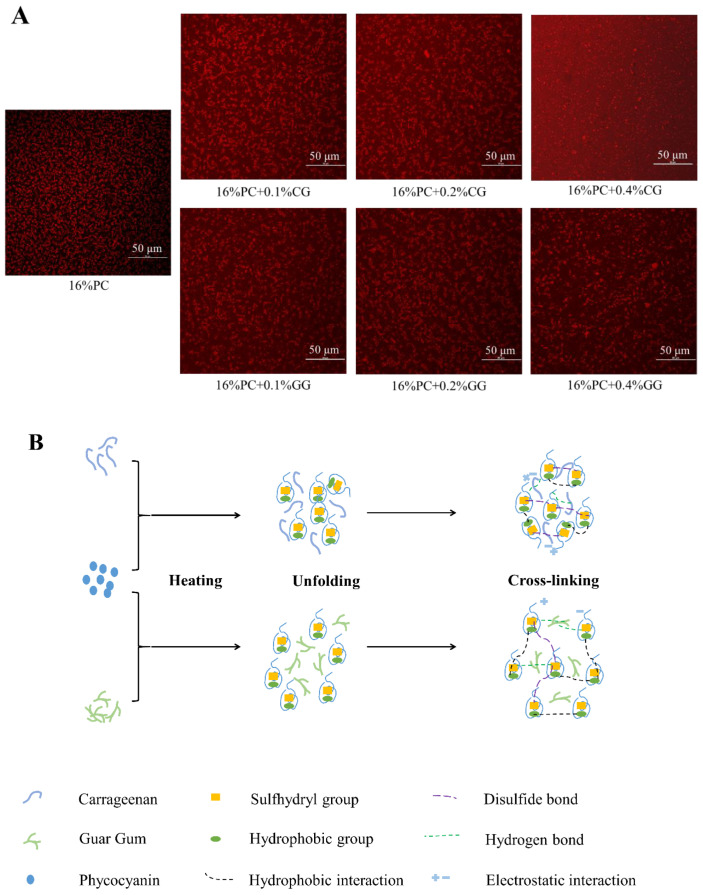
The effect of κ-carrageenan/guar gum addition on the microstructure of phycocyanin gel (**A**); (scale bar: 50 μm, red represented phycocyanin, black represented polysaccharide or water); Schematic diagram of PC-CG gels and PC-GG gels (**B**).

**Table 1 foods-11-00734-t001:** The Power Law model parameters fitting results of PC-CG and PC-GG gels at various polysaccharide concentrations.

Polysaccharide Concentration (wt.%)	Types of Polysaccharides	*G*′ = *K*′ × *ω*^*n*′^		*R* ^2^	*G*′ = *K*′ × *ω*^*n*″^		*R* ^2^
		*K*′ (Pa/s^n^)	*n*′ (Dimensionless)		*K*″ (Pa/s^n^)	*n*″ (Dimensionless)	
0	-	425.580 ± 13.064 ^Dd^	0.142 ± 0.004 ^Aa^	1.000	99.126 ± 2.723 ^Dd^	0.161 ± 0.009 ^Aa^	0.980
0.1	CG	581.754 ± 9.431 ^C^	0.140 ± 0.005 ^A^	0.998	135.276 ± 3.517 ^C^	0.142 ± 0.015 ^AB^	0.987
0.2	CG	869.311 ± 27.090 ^B^	0.129 ± 0.001 ^B^	0.999	182.768 ± 3.149 ^B^	0.129 ± 0.010 ^B^	0.985
0.4	CG	1483.286 ± 5.796 ^A^	0.091 ± 0.001 ^C^	0.998	201.776 ± 4.134 ^A^	0.119 ± 0.005 ^BC^	0.966
0.1	GG	886.015 ± 2.535 ^a^	0.144 ± 0.002 ^a^	0.999	211.169 ± 1.114 ^a^	0.143 ± 0.001 ^b^	0.989
0.2	GG	714.298 ± 19.382 ^b^	0.140 ± 0.001 ^a^	0.999	170.918 ± 6.235 ^b^	0.141 ± 0.001 ^b^	0.989
0.4	GG	481.074 ± 4.991 ^c^	0.143 ± 0.004 ^a^	0.999	115.604 ± 1.328 ^c^	0.148 ± 0.005 ^b^	0.987

Results are presented as mean values ± standard deviation of three replicates. Different upper case letters superscripted indicate significant differences among different additions of CG (*p* < 0.05). Different lower case letters superscripted indicate significant differences among different additions of GG (*p* < 0.05).

**Table 2 foods-11-00734-t002:** Critical strain (*γ_c_*), *G*′ at critical strain (*G_cr_*), and cohesive energy density (*E_c_*) of PC-CG and PC-GG gels at various polysaccharide concentrations.

Polysaccharide Concentrations (wt.%)	Types of Polysaccharides	Critical Strain *γ_c_* (%)	*G*′ at Critical Strain *G_cr_* (Pa)	Cohesive Energy Density *E_c_* (J/m^3^)
0	-	0.63109 ± 0.001 ^Cb^	508.4 ± 0.100 ^Dd^	101.241 ± 0.010 ^Dd^
0.1	CG	1.0135 ± 0.001 ^B^	707.8 ± 0.100 ^C^	363.520 ± 0.100 ^C^
0.2	CG	1.0197 ± 0.005 ^A^	1035 ± 0.050 ^B^	531.042 ± 0.005 ^B^
0.4	CG	1.0195 ± 0.001 ^A^	1663 ± 0.100 ^A^	864.245 ± 0.010 ^A^
0.1	GG	1.0248 ± 0.001 ^a^	1011 ± 0.050 ^a^	530.884 ± 0.010 ^a^
0.2	GG	0.63199 ± 0.001 ^b^	839.5 ± 0.100 ^b^	167.653 ± 0.001 ^b^
0.4	GG	0.62868 ± 0.001 ^c^	649.4 ± 0.100 ^c^	128.334 ± 0.010 ^c^

Results are presented as mean values ± standard deviation of three replicates. Different upper case letters indicate significant differences among different additions of CG (*p* < 0.05). Different lower case letters indicate significant differences among different additions of GG (*p* < 0.05).

**Table 3 foods-11-00734-t003:** The secondary structure of phycocyanin gels with different concentrations of κ-carrageenan/guar gum.

Polysaccharide Concentrations (wt.%)	Types of Polysaccharides	β-Sheet (%)	Random Coil (%)	α-Helix (%)	β-Turn (%)
0	-	25.46 ± 0.01 ^Bb^	12.35 ± 0.02 ^Aa^	13.89 ± 0.01 ^Ab^	48.30 ± 0.01 ^Db^
0.1	CG	24.89 ± 0.01 ^C^	12.05 ± 0.01 ^B^	13.63 ± 0.02 ^B^	49.43 ± 0.01 ^B^
0.2	CG	24.78 ± 0.01 ^D^	11.81 ± 0.02 ^D^	13.50 ± 0.01 ^D^	49.91 ± 0.03 ^A^
0.4	CG	25.70 ± 0.01 ^A^	11.98 ± 0.03 ^C^	13.55 ± 0.01 ^C^	48.77 ± 0.02 ^C^
0.1	GG	25.76 ± 0.02 ^a^	12.23 ± 0.01 ^b^	13.75 ± 0.01 ^d^	48.26 ± 0.01 ^c^
0.2	GG	24.96 ± 0.03 ^c^	12.03 ± 0.01 ^c^	13.87 ± 0.02 ^c^	49.14 ± 0.01 ^a^
0.4	GG	24.44 ± 0.01 ^d^	11.48 ± 0.01 ^d^	23.92 ± 0.03 ^a^	40.16 ± 0.01 ^d^

Results are presented as mean values ± standard deviation of three replicates. Different upper case letters indicate significant differences among different additions of CG (*p* < 0.05). Different lower case letters indicate significant differences among different additions of GG (*p* < 0.05).

## Data Availability

Data is contained within the article.

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
