# Peer review of "Effects of κ-Carrageenan and Guar Gum on the Rheological Properties and Microstructure of Phycocyanin Gel"

_foods, 2022, doi:10.3390/foods11050734_

Round 1

Reviewer 1 Report

The authors in this submission (Manuscript-ID foods-1579446) investigate experimentally the effect of adding the anionic polysaccharides κ-carrageenan (CG) and the neutral polysaccharides guar gum (GG) in phycocyanin (PC) protein aqueous gels. This was implemented at three low polysaccharide concentrations, up to 0.4 % w/v, for CG and GG. The in situ gelified samples were subjected to linear and nonlinear rheological tests along with additional chemical interaction forces analysis, e.g., FTIR spectroscopy, and typical confocal laser scanning microscopy, to visualize the respected microstructural conformations. Overall, the submission is based on an extensive research protocol and aims to document and reveal the behavior of the particular binary protein/polysaccharides mixtures. However, a few concerns and improvements are listed below.

The authors should consider the following remarks and improvements:

Significant concerns:

  • In Line 97, the authors write of sudden heating at 90 oC for 30 minutes, followed by zero degrees rest for 10 minutes. What was the purpose of these steps before the corresponding thermal treatment and the subsequent rheological test at 25 oC?
  • Lines 235 to 239: The description and discussion need improvement to clarify whether it is for the same group (CG family) or between the groups (CG vs. GG). Moreover, attention must be placed to avoid confusion of the reported values (the values in the parenthesis).
  • There is a slight confusion in the discussion between lines 257 to 267. The viscous decomposition dashed line (or lines) cannot be a straight line above 50% as there is an obvious intracycle variation of viscous nonlinear contribution. Especially, the comment in Line 263 does not stand, as both the total (continuous black line) and the decomposed (dashed black line) are curves. It is a matter of scales to obtain it, and I believe if the authors give a zoom-up plot of the PC Lissajous 100% strain (Figure 2B), the lines will be curved.
  • The authors should refine the text and change the expression in Lines 267 and 468 concerning the idea of “…elastic Lissajous curves ...smaller than the viscous…”. Smaller in which regard, e.g., as included area or magnitude?
  • The threshold of 50% strain is somehow subjective. The sensitive I3/I1 ratio shows a much earlier appearance of nonlinearity; for example, if one considers a 5% increase, it is at about 20% strain. Moreover, the Chebyshev coefficients ratios analysis (sufficiently developed in the text) distinguishes between the nature of nonlinearities, i.e., viscous or elastic, which is the weakness of the I3/I1 ratio. In my opinion, the viscous contribution is significant, as the Chebyshev coefficients ratios v3/v1 show 10%-15% change (Figure C1 and C2) at about 20% strain.
  • In addition, the last point at 100% strain of the e3/e1 plot disorients the reader. Consider changing the scale of Figure B1, as then, the gradual increase will also be evident in the plot of the e3/e1 ratio.
  • The authors should distinguish the intra-cycling behavior from the inter-cycling nonlinear response. The strain-stiffening is an exclusively intra-cycling structural condition and different from the overall Type I, which correctly the authors declare as thinning, but this is inter-cycling thinning. This will help the unfamiliar readers to follow the text easily.
  • Line 445: “…The pores of the gel tended to increase…” and following in lines 451 to 453. It would be helpful for the reader and easier to understand by referring to larger dimensions of the aggregates, at least as it occurs from 16%PC to 16%PC+0.4%GG. Also, refer to the basic scale in these images (50 microns) for a rough quantitative approach.
  • In Conclusions, the extensive description in Lines 470 to 477 can be reduced by qualitatively writing the basic findings. The details are already available in the Sections 3.1.2.3 and 4.
  • Finally, the message in Line 485, that the CLSM results confirm the rheological results, is vague. CLSM gives static structural info, while LAOS represents continuous transitional behavior (either as intra-cycling or inter-cycling).

General Improvements

  • In Line 70, the authors write: “…there was few research on the nonlinear rheological properties … phycocyanin-polysaccharide gels”. Either give the corresponding references or remove the remark.
  • The term LVR is introduced two times, at Lines 216 and 254, and not in the Introduction as one might expect (e.g., Line 63). Additionally, the Introduction will be shortened.
  • In Tables 1 and 2, the first columns have a different format, although they refer to the same parameter (wt.%). Keep the format consistent. In the same context, the obtained behavior is reported either as “Type I behavior” (Abstract) or as “type I behavior” (Lines 223 and 464). As it is a term, the first format is the correct one.
  • Improve Figure 2 caption. There is an unnecessary repetition of the colors’ assignment (Lines 278 and 279).
  • Line 310: instead of “symbols” use “signs”, which refers to the + or – of the Chebyshev coefficients ratios.
  • Heading of 3.5 Section: why two times the term “confocal”?
  • Consistency of the basic terms: Lines 471, 473, and 477 (possibly in other instances), as “strain stiffening” and “strain-stiffening”, and correspondingly “shear thinning” or “shear-thinning”

Language

Line 19: the repeated verb “exhibited” can be omitted in the second instance.

Lines 53-54 and 56-57: the repeating sentence structure weakens the message; the section can be improved.

Line 74: change the noun with the verb, i.e., change hypothesis to hypothesize.

Lines: 154 to 159: the section must be improved because it sounds like a technical draft for implementing the experiment/technique.

Line 175: “…all samples exhibit a gel-like network structure connected.” The language needs improvement, e.g., simplify the last part as "a gel-like structure".

Line 221: Change “…of complex fluid” to “…of a complex fluid”

Reviewer 2 Report

The manuscript works on the application of phycocyanin as a pigment along with some polysaccharides, i.e., k-carrageenan and guar gum. They have also evaluated the non-linear region of the rheological properties of the obtained gel from these polysaccharides and related their rheological properties with their microstructure. However, the manuscript prepares in a good manner, the k-carrageenan is mostly used as a gelling agent in food formula and guar gum as a thickener. Thus, what is the reason for comparing two polysaccharides in the paper? On the other side, it is not possible to compare two different things with different properties in a paper. Apart from this, there are some major concerns that can be improved:

 Line 29: There is no proper relation between the sentences. For instance, “Proteins generally have polar and non-polar groups, so they have good solubility, emulsification, gelation, and foaming properties.” Protein is amphiphilic, but it does not relate to solubility or gelation, however, it is correct for emulsification and foaming.

Line 31-33: There is also an incorrect statement.

Line 33-34: How did you reach to this that they can change food taste, structure, texture, and stability by the aforementioned statements?

Line 53: How Rheological properties can provide valuable information for understanding the microstructure of complex fluids?

Line 69-70: Please include recent works on the rheological properties of k-carrageenan and guar gum in food formulation.

Line 84: please include more information about the polysaccharide (CGN and guar gum). Please see the above manuscript for more information. The molecular weight of CG and GG are necessary.

Section 2.2. More clarification is required for sample preparations such as at which rpm did you dissolve the PC and temperature?

Line 96: How did you adjust pH to 7.0?

Line 97-98: Bathed is incorrect, please correct it. Why did the authors apply 90 oC?

Section 2.3. What is the reason for selecting these temperatures? Please refer to a previous works.

Line 113-115: Please check again the dimensions, it seems incorrect.

Line 120: what things can be found from cohesive energy density?

Line 131: solvents should be introduced in the M&M section?

Line 142: which city?

Line 148: incomplete sentence.

Line 153: The specification of the CLSM is required.

Line 162: which image processing software did you use?

Fig1 : change the x-axis to Hz and shorten the range to show better the graph. It is also recommended to use blank symbol for G’’, then it is easier to see the difference.

Line 178-180: how did you find the slope?

Line 180: It is obvious by increasing the concentration, G’ and G’’ increased, but why do you say only as CG  concentrations increased, both G' and G"?

Line 184-187: above the pI of protein, pH=7, both biopolymers have negative charge density and no complexes or aggregates were formed. For this statement, zeta-potential and particle size analyzer is required. Furthermore, optical density measurements are necessary and finally, I did not find the relation of this sentence to your work.

Line 189-191: I did not see this in Fig.1as GG concentrations increased, the G' and G" of the hybrid gel raised firstly, reached a peak when the addition amount of GG was 0.1%, and then gradually decreased”.

Line 206: dependence of phycocyanin thermal gel??

Line 206: what is the meaning of the effect of κ-carrageenan is better?

Table 1: it is better to separate CG from GG, then it is easier to compare. It is the same for Table 2.

Line 234: I am not agreeing with different types of polysaccharides, since you only use 2 polysaccharides, please consider it throughout the whole manuscript.

Line 237-240: for which samples?

 Fig. 3. Please use the shorter range for the x-axis and correct the Chinese axis.

Line 357-358: how this is possible?

Line 396-397: how did you analyzed it? By which method?

Table 3. How did the authors determine secondary structures by FTIR? Please explain?

Line 465: you did not measure more than 0.4%, then it may be possible more than0.4% CG increases the gel strength.  

Reviewer 3 Report

The Yu-chen Lei manuscript is about gels based on (k-carrageenan), guar gum and phycocyanin. The authors of the work carried out a wide range of dynamic studies of gels. Based on the data of rheological characteristics, the authors describe the structure of the systems under study. In general, the work leaves a positive impression and is done neatly. The presented results are well supported by references to already known results. The following are minor comments and questions:

Lines 28, 29. "Proteins and polysaccharides are two essential biological macromolecules in the food system, and they are important components of the food system". - I recommend correcting the statement. It contains repetitions.
Line 40. In my opinion, instead of reference "6" it is better to use these references - Derkach, S.R., Voron'ko, N.G., Maklakova, A.A. et al. The rheological properties of gelatin gels containing ?-carrageenan. The role of polysaccharide. Colloid J 76, 146–152 (2014). https://doi.org/10.1134/S1061933X14020021 and SR Derkach 1 , NG Voron'ko 2 , Yu A Kuchina 2 , DS Kolotova 2 , AM Gordeeva 3 , DA Faizullin 3 , Yu A Gusev 4 , Yu F Zuev 5 , ON Makshakova. Molecular structure and properties of ?-carrageenan-gelatin gels. Carbohydr Polym. 2018. 197:66-74. DOI: 10.1016/j.carbpol.2018.05.063. These are the works of the same team under the guidance of Prof. Svetlana Derkach.
Lines 93, 95. "0," - I propose to delete.
Line 102. "1000 ?m" - in my opinion it is better to specify the gap value in mm?! And why did the authors choose such a gap size?
Line 106. What oil was used (viscosity)?
Line 110. "(LVR)" - decryption (linear viscoelastic region) must be entered at the first mention.
Line 113. Suggest to remove - " G represents the energy storage modulus (Pa), G" represents the loss modulus (Pa)"
Lines 177-180. What caused such a difference?
Lines 209, 230. 244 et seq. "(CG or GG, 0%-0.4%)" - delete.
Fig. 1. Increase the font size.
Line 233 onwards. the authors introduce the notation Gcr and Ec. Further, you can use only them without repeated explanation. This will simplify the text of the manuscript.
Fig. 2. Designation of curves is not informative.
Lines 417-422. It was shown above that the number of hydrogen bonds decreases with increasing α-carrageenan.
Line 482. "blank control" - I suggest replacing it with "control sample" 

Round 2

Reviewer 2 Report

The authors replied to my questions properly and corrected them based on my comments. However, I've seen only my comments and no further reviewer comments was in the file.